# Linking Souls to Humans: Blockchain Accounts with Credible Anonymity for Web 3.0 Decentralized Identity

## Abstract

A decentralized identity system that can provide users with self-sovereign digital identities to facilitate complete control over their own data is paramount to Web 3.0. The account system on blockchain is an ideal archetype for realizing Web 3.0 decentralized identity. However, a disadvantage of such completely anonymous identity system is that users can create multiple accounts without authentication to obfuscate their activities on the blockchain. In particular, the current anonymous blockchain account system cannot accurately register the social relationships and interactions between real human users, given the amorphous mappings between users and blockchain identities. This work proposes zkBID, a zero-knowledge blockchain-account-based Web 3.0 decentralized identity scheme, to overcome endemic mistrust in blockchain account systems. zkBID links souls (blockchain accounts) to humans (users' personhood credentials) in a one-to-one manner to truly reflect the social relationships and interactions between humans on the blockchain. zkBID conceals the one-to-one relationships between blockchain accounts and users' personhood credentials for privacy protection using zero-knowledge proofs and linkable ring signatures. Thus, with zkBID, the users' blockchain accounts are credibly anonymous. Importantly, zkBID is fully decentralized: all user-related data are generated by users and verified by smart contracts on the blockchain. We implemented zkBID and built a blockchain test network for evaluation purposes. Our tests demonstrate the effectiveness of zkBID and suggest proper ways to configure zkBID system parameters.

## CCS Concepts

• **Security and privacy → Pseudonymity, anonymity and untraceability**; **Privacy-preserving protocols**; **Privacy protections**; *Social aspects of security and privacy*.

## Keywords

Web 3.0 Identity, Blockchain Accounts, Zero-knowledge Proofs, Linkable Ring Signatures

## 1 Introduction

Digital identity is the basis for users to manage their digital assets and interact with other users in the digital world [Windley 2005]. The development of digital identity has gone through three stages: centralized identity, alliance identity, and decentralized identity. Centralized identities are created by users but managed by application providers. One user has to endure the *complexity* of managing multiple identities and *data-leak vulnerabilities* in the multitude of applications; Alliance identities are created by users but managed by one or more large application providers. After obtaining authorization from these large application providers, the identity can be used to log into other applications without registering with the other service providers. user data are in the hands of a few large application providers, which will lead to the *data hegemony*; Decentralized identities are created and managed by users and are thus *self-sovereign*. Users maintain their identity information and, when necessary, present a self-generated proof to each application for identity confirmation. It is also widely taken that centralized identity, alliance identity, and decentralized identity correspond to the three eras of the Internet, i.e., Web 1.0, Web 2.0, and Web 3.0.

What makes Web 3.0 a revolutionary transformation is that users hold their data sovereignty, and the data is not appropriated by application providers, as in Web 1.0 and Web 2.0. A decentralized identity system with users' self-sovereign digital identities is paramount to Web 3.0. Currently, the blockchain account system serves as an archetype for a Web 3.0 decentralized identity system [Wang et al. 2023].

On blockchain, users create and manage accounts on their own, as self-sovereign identities, allowing them to handle digital assets and interact with each other. The blockchain identity system has the *anonymity* feature: nobody knows the true owner of the account, since users can create accounts without registering with a central agent. However, a large part of the economic value of activities on blockchain comes from the relationships and interactions between humans. As the anonymity, the interactions of these identities with other users cannot be traced back to the single user in a simple manner. In particular, the social interactions and relationships between real users are obfuscated. To overcome it, Vitalik Buterin proposed a conception of blockchain accounts as "soul" and a non-transferable NFT called "soul-bound" tokens to build social relationships among the accounts on the blockchain in [Weyl et al. 2022]. They illustrated how non-transferable soul-bound tokens could represent the commitments, credentials, and affiliations of "souls" and thus can encode the trust networks with economic activities to establish provenance and reputation. However, a user can still create multiple souls (accounts) on the blockchain to erase, transfer or hide relationships, and thus the blockchain account system with soul-bound tokens still cannot reflect genuine relationships in human societies.

This paper proposes a Zero-Knowledge Blockchain-account-based Web 3.0 decentralized IDentity scheme, named zkBID, to overcome the drawbacks above. With zkBID, we can link souls (blockchain accounts) to humans (users) in a one-to-one manner to truly reflect the interactions and societal relationships of humans on the blockchain. The mappings by zkBID between users and accounts are i) *decentralized* with no involvement of a third-party central agent; ii) *privacy-preserving* in that which identities are tied to which accounts are hidden (i.e., anonymous); and iii) *credible* in that each account's credit on the blockchain will be mapped one-to-one with the corresponding real-world user. Therefore, the accounts created by zkBID are credibly anonymous on the blockchain after the mapping process of zkBID.

This paper exploits zero-knowledge proofs, linkable signatures, and blockchain smart contracts to construct zkBID. Our contributions and approaches are summarized below:

1) We put forth a user-identity verification scheme using zero-knowledge proofs in zkBID. We design the Personhood credentials (PHCs) in the form of verifiable credentials and use them as digital certificates claiming the one-to-one association between users and real people identities. Moreover, we use a zero-knowledge proof algorithm to generate proofs for the validity of PHCs on the user side. The zero-knowledge proof enables a smart contract to verify the identity's validity and the identity's uniqueness without revealing the user's identity information in PHC to the verifying smart contract.

2) We construct the association of an verified user with their blockchain account using a linkable ring signature in zkBID. That is, the association maps the user to a blockchain account. A property of the linkable ring signature is that it can conceal the signer (the user). Specifically, it obfuscates the association so that, to others, a blockchain account is associated with a large set of users rather than the signing user. The association with the signer is invisible but can be verified by a smart contract on the blockchain, and thus our approach is privacy-preserving and the accounts after mapping are still anonymous. We also utilize the linkability of the linkable ring signature to ensure the uniqueness of this association (i.e., the one-to-one association between verified identities and accounts).

3) We implement zkBID in Python, Solidity and Go and deploy it on a blockchain test network of Ethereum. We conduct functional and performance tests for zkBID over the blockchain test network and analyze the results. Our tests demonstrate zkBID's effectiveness and suggest proper ways to configure zkBID system parameters.

In a nutshell, zkBID enables credibly anonymous identity-account mappings on blockchain. It is fully decentralized since all information is generated by the users themselves and verified by the smart contracts on the blockchain.

The remainder of this paper is organized as follows. Section 2 provides the preliminaries. Section 3 presents the overview of the zkBID's framework and the design details of zkBID. Section 4 analyses the security of the zkBID. Section 5 delves into the system test. Section 6 provides the discussion about related work. Section 7 concludes this paper.

## 2 Preliminaries

This section presents the technique preliminaries used to build our zkBID.

### 2.1 Verifiable Credentials and Personhood Credentials

Decentralized Identifiers (DIDs) proposed by the World Wide Web Consortium (W3C) are a new type of identifier that allows verifiable decentralized digital identity [W3C 2020]. Verifiable Credentials (VCs) play a vital role in the utilization of DID [W3C 2024]. Within the realm of VC, three primary roles come into play: the Issuer, the Holder, and the Verifier. A Verifiable Credential is a statement made by an Issuer regarding the Holder of the VC. The term "verifiable" in VC signifies its credibility and integrity, as it is securely signed by the Issuer using cryptography, allowing verification by a Verifier. The VC specification outlines its data model as a JSON object, encompassing metadata (such as the Issuer's DID, issuance date, and claim type), claims (comprising one or more assertions about the Holder, including the Holder's DID), and proofs (typically the Issuer's digital signatures).

There is an increasing concern that AIs are indistinguishable from people online. Therefore, "personhood credentials" (PHCs) [Adler et al. 2024], as a kind of digital credentials, have recently been proposed to certify that users are real people, not AIs, in online services. The operating of a PHC system is divided into two processes, i.e., the enrollment process and the usage process. In the enrollment process, a user is asked to prove that he is a real person to a PHC issuer; the PHC issuer then issues a PHC to the user to certify that the PHC issuer believes that the user applying a PHC is a real person who has not received a PHC from them previously. PHC issuers can be a range of trusted institutions, such as governments, large companies or otherwise. In the usage process, a third-party service provider can request evidence that a user holds a PHC as part of some authorization process. PHC issusers can ensure that a specific user who has not previously received a credential is a real person from them through the following three main methods:

1) Existing Identity Documents: An issuer can choose a verification method that originates from a trusted source. For instance, non-governmental issuers can issue PHCs based on government-issued identity documents rather than designing their own methods.

2) Biometric Information: By measuring persistent and unique parts of a person (such as palms, irises, or fingerprints), issuers can ensure that the user possessing human characteristics is actually a real person rather than machine, and each user is limited to registering only once.

3) Web of Trust: This method relies on social graph analysis to distinguish the real person user and machine user, and detect whether the user has received a credential before.

In the design of zkBID, we propose using the PHCs in the form of VC as the identity credential of each human user (see Section 3).

### 2.2 zkSNARK

Zero-knowledge proofs are a cryptographic technique that proves a statement's validity without revealing any private information about the statement. There are several types of zero-knowledge proof algorithms. Among them, zero-knowledge succinct non-interactive argument of knowledge (zkSNARK) is considered to be the most practical, due to their succinct and non-interactive natures [Sun et al. 2021]. First, the generated proof has a size of just several bytes, and the proof can be verified in a short running time. Second, the prover and verifier do not need to communicate synchronously with each other during both proof generation and verification.

A zkSNARK algorithm is usually represented by an arithmetic circuit that consists of the basic arithmetic operations of addition, subtraction, multiplication, and division. An $\mathbb{F}$-arithmetic circuit is a circuit in which all inputs and all outputs are elements in

a field $\mathbb{F}$. Consider an $\mathbb{F}$-arithmetic circuit $C$ that has an input $x \in \mathbb{F}^n$, an auxiliary input $w \in \mathbb{F}^h$ called a witness, and an output $C(x, W) \in \mathbb{F}^l$, where $n, h, l$ are the dimensions of the input, auxiliary input, and output, respectively. The arithmetic circuit satisfiability problem of the $\mathbb{F}$-arithmetic circuit $C$ is captured by the relation: $\mathcal{R}_C = \{(x, W) \in \mathbb{F}^n \times \mathbb{F}^h : C(x, W) = 0^l\}$, and its expression is $\mathcal{L}_C = \{x \in \mathbb{F}^n : \exists W \in \mathbb{F}^h, \text{ s.t. } C(x, W) = 0^l\}$. A zkSNARK algorithm consists of three algorithmic components [Sasson et al. 2014]:

- ■ $(PK, VK) \leftarrow KEYGEN\left(1^\lambda, C\right)$ is the key generation algorithm that generates the proving key $PK$ and the verification key $VK$ by using a predefined security parameter $\lambda$ and an $\mathbb{F}$-arithmetic circuit $C$.
- ■ $\pi \leftarrow PROVE(PK, x, W)$ is the proof generation algorithm that generates a proof $\pi$ based on the proving key $PK$, the input $x$, and the witness $W$.
- ■ $1/0 \leftarrow VERIFY(VK, x, \pi)$ is the proof verification algorithm that outputs a decision to accept or reject $\pi$ using $VK$, $x$ and $\pi$ as the input.

In this work, we employ Groth16 [Groth 2016] as the zkSNARK algrithm of zkBID for its fast verification process and compact proofs. These properties are crucial for large-scale applications and suitable for large blockchain networks.

## 2.3 Linkable Ring Signature

The concept of ring signature was introduced by Rivest [Rivest et al. 2001]. A ring signature scheme allows the verifier to ascertain that the signature was genuinely created by one of the members within this predefined set, yet it does not provide sufficient information to determine which specific member authored the signature. Compared to classical ring signature, a linkable ring signature additionally allows any verifier to ascertain the fact that two signatures were generated by the same signer [Liu and Wong 2005]. Therefor, a linkable ring signature scheme includes four algorithmic components $(GEN, SIG, VER, LINK)$ defined as follows:

- ■ $(pk, sk) \leftarrow GEN\left(1^k\right)$ is the key generation algorithm that takes security parameter $k$ as the input and outputs a public key $pk$ and a private key $sk$.
- ■ $\sigma \leftarrow SIG\left(1^k, 1^n, m, L, sk\right)$ is the signing algorithm that takes security parameter $k$, ring size $n$, private key $sk$, a ring of $n$ public keys $L = \{pk_i \mid i = 1, 2, \cdots, n\}$ including the signer's own public key (i.e., $pk_i = pk$ for some $i$), and message $m$ as the input and outputs a signature $\sigma$.
- ■ $1/0 \leftarrow VER\left(1^k, 1^n, m, L, \sigma\right)$ is the signature verification algorithm that takes security parameter $k$, ring size $n$, a ring of $n$ public keys $L$, message $m$, and signature $\sigma$ as the input and outputs 1 or 0 to indicate accept or reject respectively.
- ■ $1/0 \leftarrow LINK\left(1^k, 1^n, m_1, m_2, \sigma_1, \sigma_2, L_1, L_2\right)$ is the linkability check algorithm that takes parameter $k$, ring size $n$, two different rings of $n$ public keys $L_1, L_2$, two messages $m_1, m_2$, two signatures $\sigma_1, \sigma_2$, such that $VER\left(1^k, 1^n, m_1, L_1, \sigma_1\right) = 1$,

$VER\left(1^k, 1^n, m_2, L_2, \sigma_2\right) = 1$, as the input, and returns 1 or 0 for linked or unlinked, respectively.

A linkable ring signature scheme satisfies the properties of correctness, unforgeability, signer ambiguity, and linkability [Noether et al. 2016; Wang et al. 2008]. We will explain the properties of signer ambiguity and linkability here, as they are relevant to our design goals. Signer Ambiguity: Given a valid signature, the probability that any attacker can correctly guess the real signer, when the attacker has $t$ private keys of the ring $L$ and there are $n$ public keys in the $L$, is not greater than $1/(n - t)$. Linkability: If a signer (with the same private key) signs messages $m_1$ and $m_2$, producing signatures $\sigma_1$ and $\sigma_2$, any verifier can verify if $\sigma_1$ and $\sigma_2$ were signed by the same signer.

In this paper, we use a linkable ring signature scheme named Multilayered Linkable Spontaneous Anonymous Group Signature (MLSAGS) proposed by Noether in [Noether et al. 2016]. Compared to other algorithms, MLSAGS has a lower technical complexity in achieving linkability, making it suitable for implementing on smart contracts. In MLSAGS, a key image $y_0$ is computed from the private key and public key of the signer as $y_0 \leftarrow sk * H_p(pk)$, where $*$ is the multiplication operator of finite-field polynomials, $H_p$ is a deterministic hash function that maps a point to another point on the elliptic curve. Since the key image $y_0$ is unique for a signer with the pair of $(sk, pk)$, and all the signatures produced by the same signer are prepended with the same key image, these signatures by the same signer are linkable even if the selected public keys in the ring change.

## 3 Scheme Design

We exploit the zkSNARK algorithm, the linkable ring signature, and smart contracts of blockchain to realize the scheme design of zkBID. This section first overviews the overall design, and then gives the details on the design.

## 3.1 Design Overview

The design goal of zkBID is to connect each user's real-person identity credential to a unique blockchain account on the Ethereum blockchain, ensuring the confidentiality of the credential information and its association with the blockchain account. As a result, these blockchain accounts are deemed credibly anonymous, and are called "soul accounts".

We leverage PHCs issued by some trusted institutions (such as governments and large companies) as the users' real-person identities, and design a corresponding arithmetic circuit of zkSNARK to generate ZK proofs from the PHCs as the registration information. By storing solely the ZK proofs of the PHCs on the blockchain rather than the exact contents of the PHCs, user identity privacy is preserved while still enabling verification.

We link the hash of each verified PHC to a unique blockchain account (i.e., the soul account) in a one-to-one manner. In order to hide the one-to-one correspondence between the PHC hash and the soul account, we use linkable ring signatures to certify accounts with credible anonymity. First, we associate the hash of each verified PHC hash with a seed public key on the blockchain. Then, we apply linkable ring signatures with a set of seed public keys to certify each soul account. Thus, although the association between the PHC

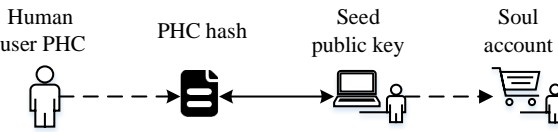

Figure 1: The mapping relationship between the user identity (the PHC and the PHC hash), and the seed public key, and the soul account.

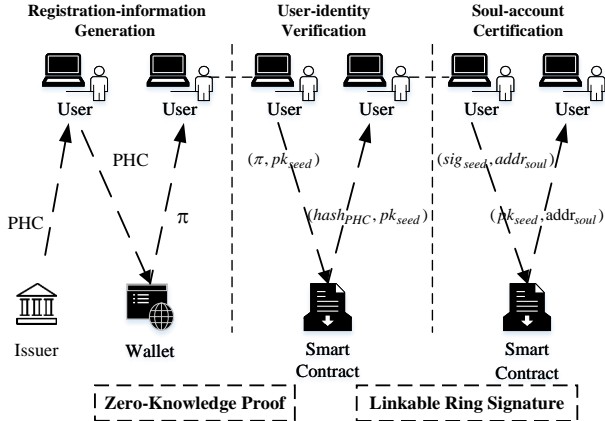

Figure 2: The overall operational flow of the IVAC process.

hash and the seed public key is visible on the blockchain, the link from the soul account to the PHC hash and the seed public key is invisible. Fig. 1 illustrates the mapping relationship between the user identity (the PHC and the PHC hash), and the seed public key, and the soul account, where the dotted line represents the on-chain invisible relationship and the arrow represents the one-way computation relationship.

zkBID generates the soul accounts with credible anonymity for users via the process that fulfills privacy-preserving identity information generation, identity verification, and identity-to-account mapping over the Ethereum blockchain. We refer to this process as the on-chain identity verification and account certification (IVAC) process. Fig. 2 illustrates the operational flow of the IVAC process. We give the design details of the three sub-processes of IVAC in the following.

## 3.2 Registration-information Generation

The primary function of the registration-information generation sub-process is to generate the registration information that will be used in the next sub-process. In zkBID, we leverage PHCs issued by some trusted institutions (such as governments and large companies) as the users' real-person identities, and design a corresponding arithmetic circuit of zkSNARK to generate zero-knowledge proof from the PHC as the privacy-preserving registration information without disclosing the exact contents contained in the PHC.

PHCs are a technical framework, in which the issuer can issue a digital credential to certify that the holder of this credential

```
{
    "id":    " http://example.edu/credentials/27",
    "issuer": "id: PHC: beedf671ebcc456e12ec62",
    "type": ["PersonhoodCredentail],
    "issuanceDate": "2024-09-14 T02: 22: 00Z",
    "expirationDate": "2024-12-14 T02: 22:00Z",
    "credentialSubject": {
        "id": "id: PHC: ebafc1f671ebbc278r16ce12",
        "type": "EdDSAVerificationKey"
        "publicKeyMultibase": "zH3C2AVvLMv6gmMNam3uVAjZpfkcJCwDwnZn6z3wXmqPV"
        }
    "proof": {
        "type":   "EdDSAVerificationKey",
        "publicKeyMultibase": "cl3l2ATLLmv6gmLSam3uVAjZplishwDwnZn76leXxpPV",
        "signature": "......"
        }
}
```

Figure 3: An example of PHC designed in the form of VC.

is a real person and not previously registered. In zkBID, we design the format of PHCs in the form of VC, and an example is shown in Fig. 3. A PHC in zkBID comprises the following three significant fields: the public key of the PHC holder (i.e., the user) $pk_{holder}$: credentialSubject.publicKeyMultibase, the public key of the PHC issuer $pk_{issuer}$: proof.publicKeyMultibase, and the signature signed by the issuer over the entire PHC $sig_{issuer}$: proof.signature.

As demonstrated in Protocol 1, to ensure privacy-preserving, the user's identity information is generated from their PHC as a zero-knowledge proof using a zkSNARK program. The user locally executes the zkSNARK's proving algorithm $zkSNARK.PROVE(PK, x, W)$ to generate the proof. The public input $x$ is composed of $hash_{PHC}$, $sig_{holder}$ and $pk_{issuer}$, where $hash_{PHC}$ is the hash of the PHC, $sig_{holder}$ is the signature signed by the private key corresponding to the holder public key recorded in the PHC (i.e., $pk_{holder}$ in the PHC), and $pk_{issuer}$ is the public key of the PHC issuer. The private witness $W$ is composed of the original PHC data $PHC$. As illustrated in Fig. 4, the arithmetic circuit of the zkSNARK program checks the validity and the ownership of the PHC according to the following steps:

- Fist, the arithmetic circuit computes the hash of $PHC$ and check if $hash_{PHC}$ in the public input is equal to the computed hash to ensure that $hash_{PHC}$ is actually computed from $PHC$ in the private input $W$.
- Second, the arithmetic circuit extracts the public key of the holder $pk_{holder}$ from $PHC$, and verifies the signature $sig_{holder}$ with the public key $pk_{holder}$, to ensure that the user owns $PHC$.
- Third, the arithmetic circuit extracts the signature of the issuer $sig_{issuer}$ from $PHC$ and verifies the signature $sig_{issuer}$ with $pk_{issuer}$, to validate $PHC$.

If the above check steps all are passed, the arithmetic circuit outputs TRUE with a ZK proof $\pi$; otherwise, it outputs FALSE. With the output of TRUE, the zkSNARK proving algorithm eventually produce the ZK proof $\pi$ for that the statement of "The PHC hash $hash_{PHC}$ is computed from $PHC$, and the signature $sig_{holder}$ is generated using the private key corresponding to the public key $pk_{holder}$ contained in $PHC$, and the signature $sig_{issuer}$ is generated using the private key corresponding to the public key $pk_{issuer}$ contained in $PHC$" is true without revealing the exact data contents of $PHC$.

---

**Protocol 1** Generation of ZK proof on the user's PHC

---

**Input**: $hash_{PHC}, sig_{holder}, pk_{issuer}, PHC$
**Setup**: $(PK, VK) \leftarrow$ zkSNARK. KEYGEN$(1^\lambda, C)$
**Generate proof**:
$x \leftarrow (hash_{PHC}, sig_{holder}, pk_{issuer})$
$W \leftarrow PHC$
$\pi \leftarrow$ zkSNARK. PROVE$(PK, x, W)$
**return** : $\pi$

---

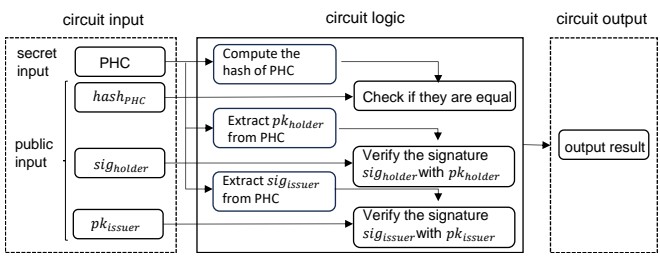

**Figure 4: The block diagram of the arithmetic circuit in the zkSNRAK algorithm.**

## 3.3  User-Identity Verification

The primary function of the user-identity verification sub-process is to verify the user's real-person identity by verifying the ZK proof of the user's identity (generated from the PHC) and to prevent the user from repeatedly registering by checking that the hash of user's PHC is not used. A user-identity verification contract on the blockchain performs these functions in this sub-process.

First, the key generation algorithm of the linkable ring signature scheme, MLSAGS, is executed to generate a key pair $(pk_{seed}, sk_{seed})$, which is called the seed key pair. Then, the ZK proof $\pi$, the public input of the zkSNARK program ($hash_{PHC}$, $sig_{holder}$ and $pk_{issuer}$) and the seed public key $pk_{seed}$ are encapsulated in the data field of a transaction, and the transaction is sent to the on-chain address of the contract to trigger the execution of the user-identity verification contract. Although the exact PHC is not sent, the contract still can verify the ownership and validity of the PHC by verifying $\pi$. And the contract prevents the user from repeatedly registering by checking that there is no more than one seed public key for the same identity (i.e., checking that the hash of the PHC is not recorded in the contract for registration previously). As demonstrated in Protocol 2, when the user-identity authentication contract is triggered by the transaction, the contract executes the following steps one by one:

- The contract gets the public key of the PHC issuer $pk_{issuer}$ predefined in the contract and checks if this predefined one is the same as the the public key of the PHC issuer contained in the public input of zkSNARK sent by the transaction.
- The contract passes the public input ($hash_{PHC}$, $sig_{holder}$ and $pk_{issuer}$) and the ZK proof $\pi$ as the input parameters to run the zkSNARK's verifying function $zkSNARK.VERIFY$.
- The contract traverse its on-chain storage space to see whether $hash_{PHC}$ has previously been recorded.

If all the steps above pass, this user is deemed a newly authenticated user. The hash of PHC $hash_{PHC}$ and the seed public key $pk_{seed}$

---

**Protocol 2** User-identity verification smart contract

---

**Input**: $\pi, hash_{PHC}, sig_{holder}, pk_{seed}$
**Authentication**:
$pk_{issuer} \leftarrow$ GetIssuerPK()
$validation \leftarrow$ zkSNARK.VERIFY$(VK, hash_{PHC}, pk_{issuer}, \pi)$
**if** $validation \neq 1$ **then**
    **return** "Verification failed"
**end if**
$stored \leftarrow$ Traverse$(hash_{PHC})$
**if** $stored \neq 0$ **then**
    **return** "Already registered"
**end if**
Store$(hash_{PHC}, pk_{seed})$
**return** : "Authentication succeed"

---

are treated as valid and they are recorded by the smart contract onto the blockchain as a key-value pair ($hash_{PHC}, pk_{seed}$). The hash $hash_{PHC}$ will be used to prevent from duplicate registration with the same PHC without disclosing the exact contents of the PHC. The seed account's public key $pk_{seed}$ recorded by the smart contract is a certificate for the user to create a privacy-preserving and credibly anonymous identity (account) over the blockchain in the soul account certification sub-process.

## 3.4  Soul Account Certification

The main function of the soul account certification sub-process is to certify that the soul account is one-to-one associated with a legal seed public key already registered on the blockchain. First, the soul account is generated as an ordinary Ethereum blockchain account. Then, the association of the soul account with the seed account is locally established by the user using the linkable ring signature algorithm. Finally, the soul account and the linkable ring signature are sent to the soul account certification contract on the blockchain to verify the established association. The verified soul account is a privacy-preserving and credible identity that enables the user to act on the blockchain in an credibly anonymous way.

For soul account certification, first, the user generates a soul account denoted by a triple that includes the account address, the public key, and the private key: ($addr_{soul}, pk_{soul}, sk_{soul}$). Then, the user selects a group of seed public keys from all current legal seed public keys, including the user's own seed public key $pk_{seed}$, to form a ring of seed public keys $L$. The user computes a key image $y_0$ based on his/her seed key pair ($pk_{seed}, sk_{seed}$): $y_0 \leftarrow sk_{seed} * H_p(pk_{seed})$. After that, the user generates a linkable ring signature in which the message to be signed is the address of the soul account $addr_{soul}$. The seed private key of the user $sk_{seed}$, the key image $y_0$ and the ring of the selected seed public keys $L$ are used to sign the message using the linkable ring signature scheme in [Noether et al. 2016]. The signing process $\sigma \leftarrow SIG(1^k, 1^n, addr_{soul}, L, sk_{seed})$ is shown in Fig. 5. Note that the produced signature has the form of $\sigma = (y_0, \cdots)$, where $y_0$ is the tag tied to the seed private key, $sk_{seed}$, to prevent multiple registrations of the same seed public key.

After that, the user constructs a transaction to trigger the user-soul-account certification contract. As shown in Protocol 3, the soul-account certification contract is executed to verify the correctness

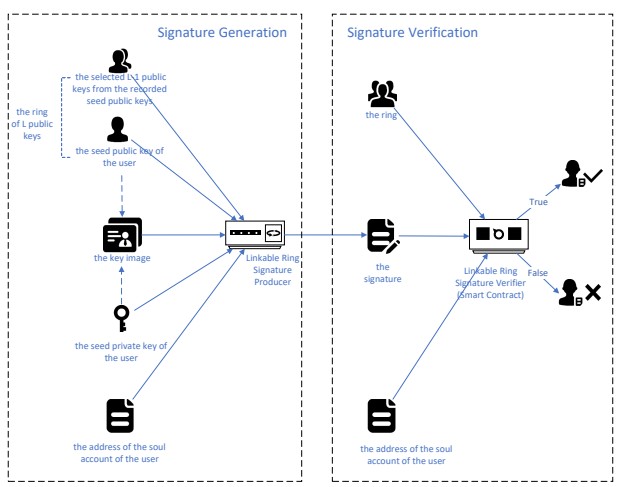

**Figure 5: The illustration of the generation and verification process of a linkable ring signature.**

and the uniqueness of the linkable ring signature $\sigma = (y_0, \cdots)$ using the signed message $addr_{soul}$ and the ring of the selected seed public keys $L$. Firstly, the smart contract will execute verification function of the linkable ring signature, $LRS.VER(1^k, 1^n, addr_{soul}, L, \sigma)$ to check if the $\sigma$ is valid. Secondly, the smart contract will check whether $\sigma.y_0$ is previously stored on the smart contract by the function $traverse(\sigma.y_0)$. If these checks are passed, the verified address of the soul account $addr_{soul}$ and the key image $y_0$ will be recorded in the on-chain storage of the soul account certification contract. In this way, the certification of the soul account, $(addr_{soul}, pk_{soul}, sk_{soul})$, is finished.

Since the key image generated by the seed public key and the seed private key is unique, one seed key pair can only generate one valid linkable ring signature for the certification, thus ensuring a strict one-to-one association of a soul account with the seed key pair. Moreover, since the association is established using linkable ring signatures, although anyone can verify the association, they cannot tell which seed public key in the ring produces this signature. Therefore, the soul account is uniquely tied to the user's identity but can still preserve privacy at the same time. The certified soul accounts can be used on the blockchain as the credible and accountable representatives of users' identities to conduct various activities.

## 4 Security Analysis

In this section, we will conduct a comprehensive analysis of the security of the zkBID protocol. The zkBID protocol has been meticulously designed to to satisfy the following security properties:

1) Identity Uniqueness: The protocol prevents malicious attackers from repetitive registration, that is, an user cannot register for multiple soul accounts.
2) Unforgeability: The protocol prevents malicious attackers from pretending an honest user to certify a new soul account.

---

**Protocol 3** User-soul-account certification smart contract

**Input**: $\sigma, L, addr_{soul}$
**Certification**:
$validation \leftarrow \text{LRS.VER}(1^k, 1^n, addr_{soul}, L, \sigma)$
**if** $validation \neq 1$ **then**
    **return** "Verification failed"
**end if**
$stored \leftarrow \text{Traverse}(\sigma.y_0)$
**if** $stored \neq 0$ **then**
    **return** "Already certified"
**end if**
$\text{Store}(\sigma.y_0, addr_{soul})$
**return** : "Certification succeed"

---

3) Anonymity: The protocol prevents malicious attackers from detecting the association between the user identity (PHC) and the soul account.

Malicious attackers may target the zkBID protocol from various dimensions. To construct a framework for the security analysis of the zkBID protocol, We set several preconditions:

- **preconditions 1:** Malicious attackers are computational adversaries with polynomial time constrains.
- **preconditions 2:** Verifiers are honest and will not collude with malicious attackers to accept false verification requests as valid ones.
- **preconditions 3:** Each user can only obtain one PHC from the issuer.

The setups of precondition 1 are commonly acceptable. Since the verifiers of zkBID protocol are smart contracts deployed on the blockchain, including the user-identity authentication and the user-soul-account certification smart contracts, we consider precondition 2 to be reasonable with the support of blockchain security. As mentioned in Section 2, issuers have various schemes to limit users from registering multiple PHCs, making precondition 3 acceptable as well. Meanwhile, the security of Groth16 and that of MLSAGS have been established in their papers [Groth 2016; Noether et al. 2016]. Therefore, we assume that the Groth16 and MLSAGS are secure. By considering these preconditions and already established securities of the cryptographic building blocks, we identify the following important attack vectors and analyse how zkBID can avoid them to maintain its security.

### 4.1 Sybil Attack

Malicious attackers, having already obtained a certified soul account, may attempt to have a new soul account using the same PHC or seed public key. For that purpose, malicious attackers have two possible methods: 1) Attackers may use the same PHC to associate with a new seed public key, then uses this new seed public key to certify a new soul account. 2) Attackers may use the seed public key that has already been used to directly certify a new soul account.

**Analysis**: We present the analysis about the above two kinds of sybil attack as follows.

1) In the process of certifying a new seed account, the user-identity verification smart contract will verify the registration information as demonstrated in Protocol 2. The verification requires that the hash of the provided PHC has not been previously stored. Due to the collision-resistance property of hash algorithms [Bakhtiari et al. 1995], each PHC can only generate one unique hash value. Furthermore, as mentioned in Section 3, the arithmetic circuit of zkSNARK will check that the PHC hash provided is indeed computed from the private input $W$ containing the whole data of the PHC. Owing to the soundness property of zkSNARKs [Groth 2016], attackers can successfully pass the verification via a falsified hash with a negligible probability.

2) When certifying a new soul account, as specified in Protocol 3, the user-soul-account certification contract checks whether the key image $y_0$ included in the input linkable ring signature has been previously used. If it has, the verification will fail. Based on the linkability of linkable ring signatures [Noether et al. 2016], attackers are unable to construct two valid signatures with different key images using the same seed private key.

Consequently, it is impossible to register two seed accounts using the same PHC or seed public key.

### 4.2 Linkage Attack

Malicious attackers can establish a linkage between the user's identity (the PHC hash) and the soul account by compromising the actual signer of the ring signature stored on the blockchain.

**Analysis**: Based on the signer ambiguity property of linkable ring signatures [Noether et al. 2016], attackers possessing $t$ private keys from the ring $L$ have only a probability of $1/(n-t)$ in correctly identifying the true signer. Hence, it becomes infeasible for attackers to establish a linkage between the user's identity and the soul account.

### 4.3 Forgery Attack

Malicious attackers may pretend honest users by forging signatures to obtain soul accounts. There are two possible methods by which attackers can achieve this goal: 1) Assuming attackers have access to an honest user's PHC, the attackers may forge the user's signature $sig_{holder}$ to generate a valid zkSNARK proof. By passing the verification of the user-identity authentication contract, the attackers can obtain a registered seed public key and use this seed public key to certify a new soul account. 2) Since the seed public keys registered of honest users are publicly stored on blockchain, if attackers can forge a valid linkable ring signature $sig_{seed}$ of an honest user, they would be able to certify a new soul account.

**Analysis**: We present the analysis about the above two kinds of forgery attack as follows.

1) According to the unforgeability property of the EdDSA algorithm [Josefsson and Liusvaara 2017] and the soundness property of zkSNARKs, attackers cannot forge $sig_{holder}$ of any honest user or create valid zkSNARK proof without valid signature.

2) According to the unforgeability property of linkable ring signatures [Noether et al. 2016], attackers are incapable of forging a valid signature of any registered seed account.

Consequently, the zkBID protocol is secure against forgery attacks.

## 5 Experimental Evaluations

In this section, we present the results of experiments conduced on our zkBID prototype system. We set up a test network for the Ethereum blockchain, consisting of six full Ethereum nodes running on the proof-of-work consensus protocol, hosted on Alibaba Cloud. Each node runs a Go-Ethereum [Ethereum Foundation [n. d.]] client. We adopted a fully connected network topology between the nodes, with each node having an independent IP address. One more user node hosted on Alibaba Cloud is connected to one Ethereum node and runs the user-side functions. All the Ethereum nodes and the user node in our experiments are equipped with an Ubuntu 20.04 operating system, an Intel(R) Core(TM) i7-10700 CPU@ 2.90GHz, and 48GB RAM.

Our zkBID prototype deploys the user-identity verification and soul-account certification smart contracts, both implemented in Solidity, on the Ethereum test network. The zkSNARK algorithm employed in this system is Groth16. The setup and proof generation for Groth16 are all accomplished on the Circom 2 platform [iden3 2024]. For linkable ring signatures, the system utilizes MLSAGS. We implemented the key generation and signing algorithms for MLSAGS in Python, while the MLSAGS signature verification contract was developed in Solidity. To assess the performance of the prototype system, we evaluated it across six key metrics:

- **ZK Proof Generation Time**: The time consumed to generate an ZKP in the Groth16 proving algorithm.
- **ZK Proof Size**: The number of bytes used to construct an ZKP in the Groth16 proving algorithm.
- **ZK Proof Verification Costs**: The amount of gas required to execute the user-identity verification smart contract on the blockchain to verify the ZKP.
- **Arithmetic Circuit Size**: The number of R1CS constraints contained in the arithmetic circuits used by Groth16.
- **MLSAGS Signature Generation Time**: The time consumed to generate a linkable ring signature of MLSAGS.
- **MLSAGS Signature Verification Costs**: The amount of gas required to execute the user-soul-account certification smart contract on the blockchain to verify a linkable ring signature of MLSAGS.

We first conducted experiments to evaluate the zkSNARK algorithm, Groth16, as applied in zkBID. Specifically, we employed a single arithmetic circuit to verify a batch of multiple PHCs and measured key performance metrics such as the ZK proof generation time, ZK proof size, ZK proof verification cost, and circuit size for different batch sizes of PHCs across varying batch sizes. As shown in Fig. 6 (a), although the circuit size increases with the batch size of PHCs, the proof size remains constant (192 Bytes), which highlights the advantage of Groth16. Fig. 6 (b) shows that both the ZK proof generation time and the ZK proof verification costs in Groth16 increase linearly with the batch size of PHCs. Due to the limitation on Ethereum's contract size, verification contracts with batch sizes of 64 and above cannot be deployed on-chain. Therefore, 6 (b) only presents the gas costs for verifying the batch of PHCs with sizes below 32. We can see from 6 (b) when a batch contains only one PHC, the proof verification costs is 215.7K gas. The proof verification costs rises to 1040.0K gas when the batch contains 32 PHCs. Moreover, we can compute that in average the

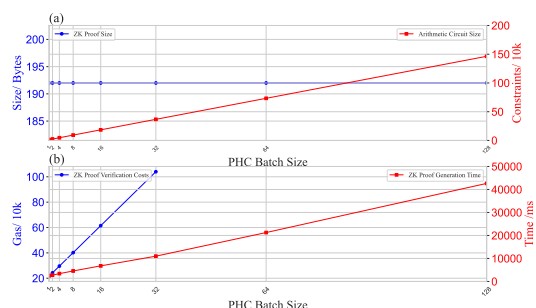

Figure 6: The costs of Groth16: (a) the number of R1CS constraints contained in the arithmetic circuits and bytes used to construct an ZKP; (b) the amount of gas required to verify the ZKP and the time consumed to generate the ZKP.

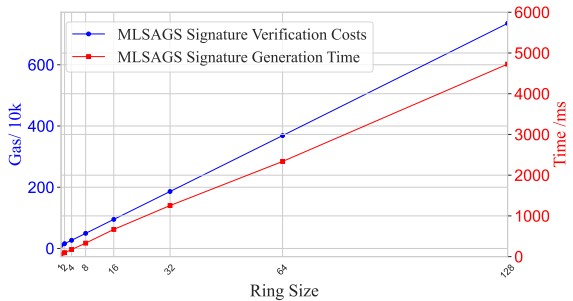

Figure 7: The amount of gas required to verify the signature of MLSAGS and the time consumed to generate the signature of MLSAGS.

gas consumption used to verify one PHC is 32.5K when the batch size is 32, which represents an approximate 7X gas consumption reduction compared to the individual verification of each PHC.

Additionally, We then conduct experiments to investigate the linkable ring signature algorithm of MLSAGS used in zkBID. Fig. 7 presents the MLSAGS signature generation time and gas consumption for on-chain signature verification with respect to different ring size $L$. Both the signature generation time and the signature verification costs exhibit a linear relationship with the ring size. For each additional seed public key in the ring, the signature generation time increases by approximately 40 milliseconds, while the signature verification cost rises by roughly 57,000 gas, both are acceptable in practice.

## 6 Related Work

There exist various approaches that leverage ZKP for privacy-preserving decentralized identity systems on the blockchain. For instance, Niya *et al.* [Niya et al. 2019] introduced TradeMap, an anonymous KYC (Know Your Customer) platform compliant with FINMA regulations, while Pauwels *et al.* [Pauwels et al. 2022] applied a ZKP-based KYC system to DeFi protocols. Aydar *et al.* [Aydar et al. 2019] and Singh *et al.* [Singh et al. 2020] proposed blockchain-based digital identity verification frameworks enabling users to access services without revealing sensitive attributes. Abraham *et*

| Work | Design Goals | | | |
|---|---|---|---|---|
| | DG1 | DG2 | DG3 | DG4 |
| [Niya et al. 2019] | ● | ○ | ○ | ○ |
| [Pauwels et al. 2022] | ● | ○ | ● | ● |
| [Aydar et al. 2019] | ○ | ● | ○ | ● |
| [Singh et al. 2020] | ○ | ● | ○ | ● |
| [Abraham et al. 2020] | ● | ● | ○ | ● |
| [Rathee et al. 2022] | ○ | ● | ○ | ○ |
| [Zhang et al. 2024] | ● | ○ | ● | ○ |
| [Kim and Ryou 2023] | ○ | ● | ○ | ○ |
| zkBID | ● | ● | ● | ● |

Table 1: Evaluating schemes against design goals.

*al.* [Abraham et al. 2020] extended the self-sovereign identity (SSI) model to support credential revocation and offline verification. In a similar vein, Rathee *et al.* [Rathee et al. 2022] introduced the ZEBRA scheme, utilizing zkSNARKS for cost-effective on-chain anonymous credential verification. Zhang *et al.* [Zhang et al. 2024] proposed a ZKP identity authentication protocol applicable to energy trading. Additionally, Kim *et al.* [Kim and Ryou 2023] presented a digital identity authentication system for the metaverse that combines Soulbound Tokens (SBTs) and Decentralized Identifiers (DIDs) for KYC processes, ensuring privacy and compliance through ZKPs. Table I offers a comparative analysis of our zkBID solution with these approaches based on the key design goals giveb below.

**DG1:** Technology Flexibility: The solution is adaptable and not limited to any specific technology, such as a particular ZKP algorithm or blockchain.

**DG2:** Identity Anonymity: No entity or individual can ascertain or derive the identity of a user from their on-chain activities.

**DG3:** Identity Credible: Each identity's can credibly be connected with the corresponding real-world user.

**DG4:** Privacy-preserving document storage: Identity documents are stored confidentially and are not publicly visible.

## 7 Conclusion

This work introduces zkBID, a decentralized Web 3.0 identity solution on the blockchain that leverages zero-knowledge proofs, linkable ring signatures, and smart contracts. By addressing anonymity challenges in current blockchain account systems while safeguarding user identity privacy, zkBID employs an identity verification framework based on zero-knowledge proofs and a mapping system using linkable ring signatures. Identity verification in zkBID utilizes the recent personhood credentials to ensure the legitimacy and uniqueness of the registered identities. Through a zero-knowledge proof algorithm, users can generate proof for their identities from personhood credentials securely. The one-to-one mapping of verified identities to blockchain accounts in zkBID is protected by linkable ring signatures, enhancing user privacy. These signatures also ensure the uniqueness of the mappings. We developed a zkBID prototype and tested it on a 6-node blockchain network on Alibaba Cloud, demonstrating its functionality and performance. Comparative analysis with existing blockchain based identity solutions highlights the distinct advantages of zkBID.

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
