# OpenReview forum: "Linking Souls to Humans: Blockchain Accounts with Credible Anonymity for Web 3.0 Decentralized Identity"
_ACM.org/TheWebConf/2025/Conference — WWW 2025 Poster_

### Official Review · Reviewer_qKKR · 2024-11-12

**Novelty:** 3
**Technical Quality:** 5

**Review:**

The paper titled "Linking Souls to Humans: Blockchain Accounts with Credible Anonymity for Web 3.0 Decentralized Identity" presents zkBID, a proposed decentralized identity solution designed for Web 3.0 applications. The main novelty lies in its approach to achieving credible anonymity by linking blockchain accounts (souls) to real-world identities (humans) using zero-knowledge proofs (zkSNARKs) and linkable ring signatures.


The paper introduces zkBID as a framework for:

1.	zkBID uses personhood credentials (PHCs) and zero-knowledge proofs to validate real-world identities on-chain without revealing personal data.


2.	It uniquely links PHCs to blockchain accounts using linkable ring signatures, ensuring that each soul account corresponds to a single real-world user.

3.	By leveraging zkSNARKs and linkable ring signatures, zkBID achieves a high degree of credible anonymity while ensuring that the identity-to-account relationship remains private.

1.	Typographical Issues:
o	"PersonhoodCredentail" (Figure 3): Should be "PersonhoodCredential".
o	Inconsistent capitalization in sections such as “user identity authentication” vs. “User Identity Authentication”.
2.	Clarity in Explanations:
o	In Section 3.2, the description of Protocol 1 would benefit from rephrasing for clarity. Specifically, the details of the zero-knowledge proof generation process could be simplified to avoid redundancy.
o	The flow diagrams (Figures 1-5) could be better annotated for improved readability. In particular, Figure 5 lacks clarity on how the MLSAGS signature process integrates with the Ethereum smart contract.
3.	Technical Missteps:

o	Inconsistent terminology: "soul accounts" and "blockchain accounts" are used interchangeably. Clarifying these terms would improve technical consistency.

o	The paper mentions Groth16 zkSNARK for “large blockchain networks,” yet Ethereum, which is known for limited throughput and high gas costs, may not fully support this. A discussion on compatibility with more scalable blockchain networks like Polygon or zk-rollup solutions would be relevant.

1.	The main elements of the zkBID proposal: Zero-knowledge proofs (zkSNARKs), personhood credentials (PHCs), and linkable ring signatures are well-established in the field of privacy-preserving digital identities. Several existing studies already employ these techniques to achieve similar goals, such as credible anonymity, verifiable user identity, and decentralized identity management. The paper's approach largely resembles current methods without introducing any major advancements in technique or theory:
2.	The paper lacks improvements or optimizations for the broader limitations found in current privacy-preserving identity systems, such as:
-	The zkBID system relies on Ethereum’s network for deployment, but this is known for high gas fees, especially when using zkSNARKs. The paper does not propose any optimization to reduce gas costs or improve efficiency, which has been a focus in other studies to make zkSNARKs more cost-effective on blockchain platforms.
-	The paper admits linear increases in gas and computation costs as batch sizes increase but does not address scalability solutions, such as employing more efficient circuits or alternative proof systems like zk-STARKs or Bulletproofs, which might reduce verification costs.
3. The framework and protocols presented here largely repurpose existing methodologies without introducing new theoretical insights. The technical components are essentially straightforward applications of zkSNARKs and linkable ring signatures, already discussed extensively in the literature on privacy-preserving blockchain identity systems:
-	There is no novel approach to improve the computational efficiency or security robustness beyond what current solutions offer.
-	The proposed system does not introduce any distinct approach to ensure that the identity remains unlinkable to multiple accounts beyond existing methods, and the security analysis is basic, with limited exploration of new attack vectors or defense mechanisms.
4. The paper does not sufficiently benchmark zkBID against other blockchain-based identity solutions. For instance, there is no clear explanation of how zkBID’s privacy, security, or efficiency metrics compare to other frameworks using zkSNARKs or ring signatures. The absence of comparative analysis weakens the paper’s claim to improve or differentiate itself from the field.

**Questions:**

How can the zkBID system be optimized to reduce gas costs and improve scalability on Ethereum or other blockchain platforms? What are the potential benefits and drawbacks of using alternative proof systems (e.g., zk-STARKs, Bulletproofs) or layer-2 scaling solutions?

What are the potential attack vectors against the zkBID system, and how can these be mitigated? Can the system be further strengthened to ensure the long-term privacy and security of user identities?

How can the zkBID system be differentiated from existing privacy-preserving identity solutions? What are the unique features or advantages that it offers over other approaches?

**Reviewer Confidence:**

3: The reviewer is confident but not certain that the evaluation is correct

**Scope:**

4: The work is relevant to the Web and to the track, and is of broad interest to the community

---

### Official Review · Reviewer_PnpP · 2024-11-27

**Novelty:** 6
**Technical Quality:** 5

**Review:**

--Overall

When proposing the zkBID scheme, the author conducted an in-depth analysis of the anonymity problem of the blockchain identity system and proposed a practical solution. The technical solution in the paper is detailed and logically rigorous. Through advanced technologies such as zero-knowledge proof and linkable ring signature, it ensures that the user's identity on the blockchain is both credible and anonymous. In addition, the author also verified the effectiveness and performance of the scheme through experiments, showing a high academic level.

The structure of the article is relatively clear. From the introduction, technical background, scheme design, security analysis to experimental evaluation, each part is naturally connected and the logic is smooth. The author uses easy-to-understand language and charts to explain complex technical concepts, so that readers can better grasp the core content of the paper.

The paper has a high degree of originality. The zkBID scheme proposed by the author is a novel solution in the field of blockchain identity system, especially in dealing with user identity anonymity and credibility. By combining technologies such as zero-knowledge proof and linkable ring signature, a one-to-one mapping between user identity and blockchain account is achieved, while protecting user privacy.

With the advent of the Web3.0 era, user data sovereignty and identity anonymity have become key issues. The zkBID solution provides a feasible technical path to solve these problems, which is of great significance for promoting the application of blockchain technology in the field of identity management. In addition, the solution also has high practical application value, and can provide a more secure and reliable identity authentication mechanism for social relationships, economic activities and other fields on the blockchain.

--pros

Through technologies such as zero-knowledge proof and linkable ring signature, the zkBID solution can verify the identity without leaking the user's identity information, and can provide users with a high degree of privacy protection.

The zkBID solution is built on the blockchain, has the characteristics of decentralization, can resist single point failures and attacks, and improves the stability and security of the system.

--cons

The zkBID solution involves advanced cryptographic technologies such as zero-knowledge proof and linkable ring signature, as well as the smart contract and consensus mechanism of the blockchain, which makes the technical implementation and maintenance of the solution relatively complex and requires high professional skills of developers.

Despite the continuous advancement of blockchain technology, it may still face performance bottlenecks when processing a large number of transactions, such as limited transaction throughput and extended transaction confirmation time. This may lead to a decline in user experience during peak hours.

Due to the anonymity and decentralization of the blockchain, the zkBID solution may face regulatory compliance challenges. How to ensure the legal use of user identity information while meeting the requirements of data protection and privacy regulations is a problem that needs to be solved.

For ordinary users, understanding and using blockchain-based identity authentication schemes may require a certain learning cost. Users need to be familiar with the new authentication process, security measures, and how to deal with potential problems.

Although the zkBID scheme has designed a variety of security measures to protect user identity and transaction security, no system can completely eliminate security risks. For example, smart contracts may have vulnerabilities and blockchain networks may be attacked, which may lead to the loss of user funds or identity information.

The operation and maintenance of blockchain networks require a lot of computing resources and energy, especially when consensus mechanisms such as proof of work (PoW) are adopted. This may lead to high operating costs and have a negative impact on the environment. Although some schemes are exploring more energy-efficient consensus mechanisms, this issue still needs attention.

**Questions:**

The components of the zkSNARK algorithm are introduced, but the technical difficulties and challenges of implementing these components are not mentioned.

Some security analysis is performed, but a comprehensive analysis of other potential attacks may be lacking.

Although six key indicators are evaluated, there may be other important performance indicators that are not considered.

The compatibility of the zkBID scheme with other blockchain technologies and protocols is not discussed in the paper.

There are inconsistencies between the theoretical description and the actual implementation details.
Some sentences are long and complex in structure, which may cause confusion to readers when reading.

**Reviewer Confidence:**

4: The reviewer is certain that the evaluation is correct and very familiar with the relevant literature

**Scope:**

4: The work is relevant to the Web and to the track, and is of broad interest to the community

---

### Official Review · Reviewer_9tbR · 2024-12-01

**Novelty:** 6
**Technical Quality:** 6

**Review:**

The paper presents a decentralized identity system built using blockchain. However, since the system allows completely anonymous
identities it needs to face the problem of sybil attacks able to create many false identity accounts on the blockchain. The paper addresses this issue presenting a defense to such an attack based on zero-knowledge proofs. The paper presents also a proof of concept of the system and a preliminary evaluation.
I like the paper's decentralized approach and the motivation of not concentrating the digital identities on a few providers but transferring their management directly to the owners. This may not be practical in some contexts but is however valuable to have the alternative to do so in others.  Furthermore, a decentralized approach is more robust and does not suffer from the single point of failure problem.

The paper uses existing cryptographic primitives to build the zkBID system. The primitives are not novel but the zkBID is and indeed the paper proposes an original contribution and a solution to an open issue.

Personhood credentials (PHC) attest that the user is a real person, not a proxy or a program. However, the decentralised nature of the issuance of PHC cannot prevent a person from getting several PHC from different PHC issuers. PHC issuers can keep track of whom they certify but there could be many PHC issuers and since they operate independently they cannot find out about users getting PHC already from other issuers. This leads to multiple legitimate digital identities.

I think this is limited by precondition 3, but this precondition is completely arbitrary and difficult to enforce in practice.

Verification of ZKP and the signatures is quite costly if you consider that the service provider must perform it to verify the user. So, this scheme can indeed create heavy loads for the service providers.

Not so sure WWW is the best fit for such a paper since the novelty relies pretty much to the use of cryptographic algorithms so a conference more oriented on crypto would be a better fit.

**Questions:**

Often cryptographic primitives and algorithms have corner cases or special cases where parameters must be chosen in a specific way. In the paper there is no limitation and no mention of any corner case, did you ignore them, or all the primitives you have chosen work as expected in all general cases? If the former, you should explicitly mention the corner cases.

How your scheme deal with the case of multiple PHC issued to the same user I described above?

**Reviewer Confidence:**

3: The reviewer is confident but not certain that the evaluation is correct

**Scope:**

4: The work is relevant to the Web and to the track, and is of broad interest to the community

---

### Official Review · Reviewer_42dx · 2024-12-02

**Novelty:** 3
**Technical Quality:** 4

**Review:**

Strengths:
This paper presents a solution to a significant challenge in Web 3.0 decentralized identity systems - establishing credible anonymity while maintaining one-to-one mappings between real users and blockchain accounts. The work introduces a novel combination of zero-knowledge proofs and linkable ring signatures to achieve privacy-preserving identity verification, develops a practical framework for linking "souls" (blockchain accounts) to human users while maintaining anonymity, and provides a comprehensive implementation and evaluation on a test network. The technical depth is impressive. The experimental evaluation provides concrete performance metrics and scalability analysis. The work builds meaningfully on existing literature while addressing an important gap in current blockchain identity systems.

Weaknesses:
1.The paper mentions using "trusted institutions" for issuing Personhood Credentials (PHCs) but doesn't fully explore the centralization risks this introduces into an otherwise decentralized system.
2. The security analysis could be strengthened by more explicitly stating privacy assumptions and potential attack vectors in the linkable ring signature scheme, particularly around ring size selection.
3. While the test implementation is valuable, the paper could benefit from more discussion of practical deployment challenges in real-world blockchain networks, including scalability limitations and integration with existing systems.
4. The paper focuses primarily on technical aspects but provides limited discussion of user experience considerations, such as the complexity of managing PHCs and interacting with the system.
5. Recovery Mechanisms: There's limited discussion of account/credential recovery mechanisms in case of key loss or compromise.

**Questions:**

Questions:
1. The evaluation shows the linear scaling of gas costs with ring size in MLSAGS. What ring size do you recommend for practical deployments to balance privacy and costs? Have you analyzed minimum ring sizes needed to achieve meaningful privacy guarantees?"
2. Could you elaborate on the trust model for PHC issuers? Specifically: What prevents collusion between multiple issuers? How do you handle the revocation of compromised issuer keys? What happens if an issuer goes offline/bankrupt?
3. The paper doesn't discuss recovery mechanisms. How would you handle scenarios like: Lost private keys for the soul account; Compromised PHC credentials; Need to transfer identity to a new blockchain account Without compromising the one-to-one mapping guarantee?
4. The linkable ring signature provides k-anonymity based on ring size, but: What additional metadata might leak through transaction patterns? Have you analyzed the system's resistance to correlation attacks across multiple transactions? What privacy guarantees remain if some ring members are compromised?
5. How would gas costs scale on mainnet Ethereum vs your test network?
6. What modifications would be needed to deploy on L2 scaling solutions?
7. How would you handle upgrades to the smart contracts while preserving existing identities?"
8. Have you analyzed compliance with privacy regulations like GDPR? Specifically: How would the 'right to be forgotten' be implemented? What user data is considered personal under GDPR? How would mandatory identity disclosure orders be handled?
9. How would zkBID integrate with: Existing DID/VC systems; Traditional identity providers
10. The system requires significant on-chain storage and computation. Who bears the gas costs for registration and verification? Have you considered incentive mechanisms for sustaining the system? How would you prevent DoS attacks through mass registrations?
11. Your results show efficiency gains with batch processing of PHCs. Could you elaborate on:
Practical limitations of batch sizes in production; Privacy implications of batching; Strategies for optimal batch composition?

**Reviewer Confidence:**

3: The reviewer is confident but not certain that the evaluation is correct

**Scope:**

3: The work is somewhat relevant to the Web and to the track, and is of narrow interest to a sub-community

---

### Official Review · Reviewer_gCiR · 2024-12-05

**Novelty:** 3
**Technical Quality:** 5

**Review:**

The paper presents a decentralized identity scheme, named zkBID, that enables one-to-one linking of users’ blockchain account (“soul”) to their real-world human identity. The proposed solution uses zero-knowledge proofs for privacy-preserving verification of human identity by a smart contract and a linkable ring signatures to associate a verified user with their blockchain account without revealing user’s identity.  The paper presents a security analysis of the zkBID scheme and its implementation on Ethereum with performance evaluation.

Pros:
- Use of linkable ring signature to allow a user to associate its blockchain identity with a verified identity while preserving the real identity is novel and interesting.
- Security analysis with well-defined threat model and attack analysis.
- An implementation with reasonable performance overhead/cost.

Cons:
- Limited novelty in the zero-knowledge proof (ZKP) for decentralized identity as other known works use ZKP for similar purposes.
- Unclear motivation and questionable adoption in practice: idea of self-sovereign identity is to enable users to leverage multiple identities for different purposes and this solution effectively breaks that premise.
- Closely-related works on decentralized identity missing.


The paper does a good job in presenting a privacy preserving solution to allow blockchain identities to be associated with real identities. While ZKP has been previously used in proving characteristics of an identity without revealing the identity attributes themselves (e.g. Sovrin [1]), the additional step of the use of linkable ring signature provides proof that a blockchain identity is associated with a real-world user identity. Overall, the security analysis is well-described and supports the formulation of the proposed scheme.

One weakness of the paper is its limited novelty. Use of ZKPs for blockchain-based identity have been explored before [1][2][3] with similar flows and use cases for verifiable credentials. The related works section is completely missing the body of work in this space and the novelty of the use of ZKP is not clear.

The premise of self-sovereign identity solutions is to provide users with full control over their identities. As a result, multiple issuers can provide different verifiable credentials to the same user; this is also typical for real-world identity issues, such as DMV or passport office. It is not clear how the proposed scheme can prevent the same user from having multiple blockchain addresses associated with different real-world identities. If thats the case, in what scenarios would the solution be useful for? If it intends to prevent one user from having multiple identities, it also breaks the premise of self-sovereign identity. To improve the overall motivation of the proposed solution, it would be useful for the paper to include specific scenarios and applications where such anonymous one-to-one linkage would be needed.


[1] https://sovrin.org/the-sovrin-network-and-zero-knowledge-proofs/

[2] https://hyperledger-fabric.readthedocs.io/en/release-1.2/idemix.html

[3] https://www.lfdecentralizedtrust.org/projects/hyperledger-indy

**Questions:**

- How does the work differ from prior works in the use of ZKP for decentralized identities?
- What are the specific scenarios where anonymous linkage between blockchain identities and real work identities would be necessary?
- The user can still have multiple blockchain addresses that are associated with different identities that they can obtain from multiple issuers (e.g. from banks, govt. agencies, private individuals, etc.). How does the proposed solution help in such a case?

**Reviewer Confidence:**

3: The reviewer is confident but not certain that the evaluation is correct

**Scope:**

3: The work is somewhat relevant to the Web and to the track, and is of narrow interest to a sub-community